# Deep Learning–Based Segmentation of *Trypanosoma cruzi* Nests in Histopathological Images

Nidiyare Hevia-Montiel [1,*], Paulina Haro [2], Leonardo Guillermo-Cordero [3] and Jorge Perez-Gonzalez [1]

1   Unidad Académica del Instituto de Investigaciones en Matemáticas Aplicadas y en Sistemas del Estado de Yucatán, Universidad Nacional Autónoma de México, Km 4.5. Carretera Mérida-Tetiz, Ucú 97357, Yucatan, Mexico; jorge.perez@iimas.unam.mx
2   Instituto de Investigaciones en Ciencias Veterinarias, Universidad Autónoma de Baja California, Mexicali 21386, Baja California, Mexico; paulina.haro@uabc.edu.mx
3   Facultad de Medicina Veterinaria y Zootecnia, Universidad Autónoma de Yucatán, Km. 15.5. Carretera Mérida-Xmatkuil, Tizapán 97100, Yucatan, Mexico; leonardo.guillermo@correo.uady.mx
*   Correspondence: nidiyare.hevia@iimas.unam.mx

**Abstract:** The use of artificial intelligence has shown good performance in the medical imaging area, in particular the deep learning methods based on convolutional neural networks for classification, detection, and/or segmentation tasks. The task addressed in this research work is the segmentation of amastigote nests from histological microphotographs in the study of *Trypanosoma cruzi* infection (Chagas disease) implementing a U-Net convolutional network architecture. For the nests' segmentation, a U-Net architecture was trained on histological images of an acute-stage murine experimental model performing a 5-fold cross-validation, while the final tests were carried out with data unseen by the U-Net from three image groups of different experimental models. During the training stage, the obtained results showed an average accuracy of $98.19 \pm 0.01$, while in the case of the final tests, an average accuracy of $99.9 \pm 0.1$ was obtained for the control group, as well as $98.8 \pm 0.9$ and $99.1 \pm 0.8$ for two infected groups; in all cases, high sensitivity and specificity were observed in the results. We can conclude that the use of a U-Net architecture proves to be a relevant tool in supporting the diagnosis and analysis of histological images for the study of Chagas disease.

**Keywords:** automatic nest segmentation; chagas disease; convolutional neural network; deep learning; histopathological imaging; *Trypanosoma cruzi* infection



## 1. Introduction

The use of artificial intelligence in the medical imaging area has been developing for several years; however, particularly in the last 10 years, the use of deep learning algorithms has shown robust results related to pattern detection and segmentation in several biomedical applications [1].

In this sense, the need to be able to automatically detect certain patterns and to segment regions of interest in images is of great importance, as for example in the images acquired through a microscope, because on many occasions this work is subjective and observer-dependent. Such is the particular case in diagnosis of imaging for histopathological studies, in which the microphotographs show various patterns or regions that have different colours, textures, and/or shapes that allow the expert to give a pertinent clinical interpretation. Histopathological images are an excellent use case for application of deep learning strategies, where a first challenge has been to analyze individual cells for accurate diagnosis from deep convolutional neural network to robustly and accurately detect and segment cells, implementing cascaded by multi-layer convolution operation without subsampling layers for cell segmentation [2]; or some cell segmentation works that tackle the data heterogeneity problem by cost-sensitive learning strategy to solve the imbalanced data distribution, and post-processing step based on the controlled watershed to alleviate fragile

cell segmentation with unclear contour [3]. However, histopathological analysis of images acquired by an optic microscope can take a lot of time to detect associated patterns, as for example in the case of parasite nests, which can result in a subjective, observer-dependent analysis. It is in facing this type of challenge that the use of algorithms derived from deep learning can be a robust tool that helps experts in the search for findings or patterns for a diagnosis [4,5].

In this work, we focus on the automatic analysis of microphotographs for the study and histopathological diagnosis of Chagas disease (CD), which is a parasitic disease caused by the flagellated protozoan *Trypanosoma cruzi* (*T. cruzi*). There are several strains of *T. cruzi*, and each one presents tropism for different cells. CD is an endemic disease in various countries in Latin America, and it is estimated that it affects around 6 to 7 million people there, in addition to 300,000 in the United States and between 80,000 and 120,000 in Europe. However, the World Health Organization (WHO) has classified CD as one of the least attended tropical diseases [6]. In Mexico, even though official records report a few cases per year, it is believed that at least 1.1 million people are infected with *T. cruzi*, 29.5 million are at risk of infection, and 30% of symptomatic infected people develop serious cardiomyopathy due to myocardial lesions, varying from mild affectations to end-stage heart failure [7].

The life cycle of *T. cruzi* is complex and presents four distinct developmental stages: epimastigotes, metacyclic trypomastigotes, bloodstream trypomastigotes, and amastigotes. The first two stages are found in the vector and the last two in the infected mammal. During an acute infection, the bloodstream trypomastigote can be found circulating in the blood and infecting cells, mainly cardiac cells. Once inside the cell, the parasite is transformed into an amastigote, the replicative form of the parasite in the host. These dividing amastigotes (via binary fission) infect cells from the phagocytic system or lymphoid, muscular, or nervous tissue, where these dividing forms continue to grow for up to nine generations inside the cell and can rupture the plasma membrane when they turn back into trypomastigotes [8]. In the chronic infection phase, the parasite is no longer found circulating in the bloodstream, and it persists in the amastigote form in cardiac cells. In 30–40% of infected and untreated patients, Chagas cardiomyopathy (CC) develops; CC can progress and cause heart failure and death. Although all stages play a role in the life cycle of the parasite, in this work we will focus on the amastigotes.

CD is characterized by *T. cruzi* amastigotes infecting myocardium tissue, as is shown in Figure 1b, in comparison with healthy tissue as in Figure 1a, which may induce inflammation of the four cardiac chambers and, later, cardiomegaly (increase of cardiac mass). Histological samples show amastigote nests accumulate inside cardiomyocytes, resulting in cardiac fibrosis, degeneration, and necrosis [9].

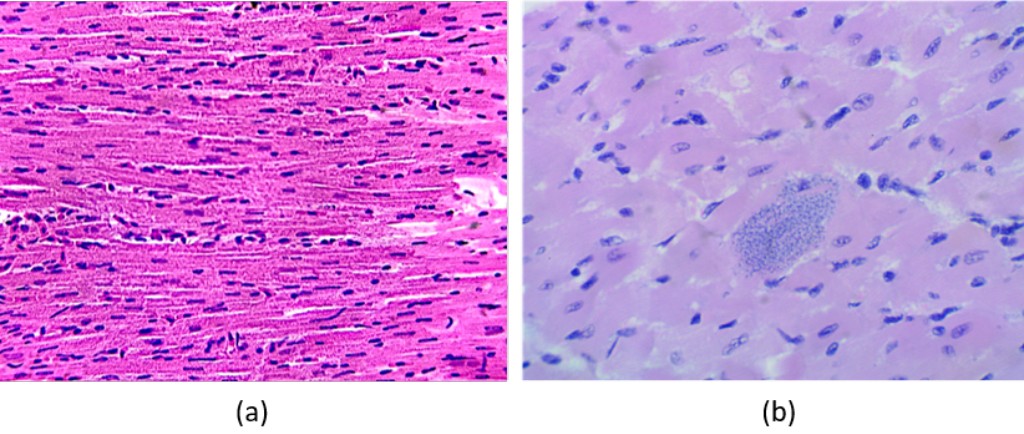

(a)       (b)

**Figure 1.** Histological samples images of cardiac tissue in the experimental model (haematoxylin and eosin stain): (**a**) control animal without *T. cruzi* infection, and (**b**) infected animal with amastigote nests of *T. cruzi* inside cardiomyocytes.

At present, the advances that exist in the study of the detection and segmentation of the *T. cruzi* parasite through the use of machine learning algorithms have only been for bloodstream trypomastigotes, that is, when the parasite is circulating in the bloodstream, as is presented in the works by Cetina et al., who proposed a *T. cruzi* automatic detection in blood smears using a Gaussian discriminant analysis (with 98.3% sensitivity and 15.6% specificity), or using a support vector machine (SVM) in combination with AdaBoost (with 100% sensitivity and 93.2% specificity) [10,11]. Recently, for the detection of trypomastigotes in the bloodstream, machine learning (ML) algorithms such as random forest (RF) have also been proposed that use images obtained by mobile phones, reporting 87.6% precision and 90.5% sensitivity [12]. Ojeda et al. [13], implemented deep learning algorithms for *T. cruzi* trypomastigote segmentation from blood samples, which obtained an F2 score of 80.0%, recall of 87.0%, precision of 63.0%, and a Dice score of 68.2%.

Some works have also reported on the detection and segmentation of parasites present in blood by implementing deep learning techniques, but related to other diseases, as for example Mehanian et al., who used convolutional neural networks (CNNs) to identify and quantify the presence of malaria parasites, Plasmodium falciparum, in blood smears (with sensitivity of 91.6%, specificity of 94.1%, and precision of 89.7%) [14]; another example is the work presented by Gorriz et al. [15], where they applied a U-Net architecture for segmentation and classification of leishmaniasis parasites' amastigotes (with precision of 75.7% and a Dice score of 77.7%). As mentioned, processing histopathological images to segment amastigotes' nests continues to be a challenge in the study of Chagas disease. Amastigote nest detection on myocardial histopathology can be helpful in the case of donors and recipients of cardiac transplants. The reactivation of infection after transplant has been reported as a serious complication that can lead to allograft failure and death. Automatic detection algorithms can also be useful in biomedical research for the study of cardiac infection in animal models and as a support diagnostic tool in places where no expert is available.

Unlike the state of the art previously reported, this work presents the following contributions:

- The implementation of a deep neural network architecture with a data augmentation strategy for the automatic segmentation of *T. cruzi* amastigotes' nests. It can help in the experimental study of Chagas disease.
- Preparation of murine experimental models during the acute infection stage, and acquisition of cardiac muscle histology images from four different experimental studies of *T. cruzi* infection.
- The proposed methodology for automatic segmentation was thoroughly validated using three sets of images (hold-out data) with different properties such as: quantity of inoculated trypomastigotes, days post-infection, acquisition equipment, and images of infected models and healthy controls.

To our knowledge, there are no current studies that do automatic amastigote nest segmentation. It is expected that the proposed methodology can help to perform an efficient and accurate detection of amastigote nests in histology images.

## 2. Materials and Methods

This section presents a brief description of the experimental murine model in the acute stage of *T. cruzi* infection, where optical microphotographs were acquired for histopathological analysis and, subsequently, the amastigote nests' segmentation by applying deep learning techniques. Likewise, the databases used for the neural network training and the final performance tests of the segmentation algorithm are presented, as well as the performance metrics used for this analysis.

### 2.1. Experimental Murine Model Description

For the experimental model of *T. cruzi* infection in the acute stage, approved by Comité de Ética del Centro Regional de Investigaciones Dr. Hideyo Noguchi de la Universidad

Autónoma de Yucatán (CIR-UADY) (CIRB-006-2017) (CEI-04-2020), 18 female ICR mice of 6–8 weeks old were considered, separated into a control group (3 mice) and an infected group (15 mice. The control mice were intraperitoneally administered with physiological saline solution, and in the infected group, nine mice were intraperitoneally administered with 1000 bloodstream trypomastigotes and the other six mice with 500 bloodstream trypomastigotes, both of H1 *T. cruzi* strain DTU I. The experimental murine model of the acute infection stage was analysed at different post-infection days (25, 28, 29, 30, 31, and 35 days post-infection).

In all cases of inoculated mice, developing infection was confirmed and monitored to corroborate the presence of *T. cruzi* parasites by microscopy using a Neubauer cell count of peripheral blood parasites every post-infection test day, and mortality was recorded daily. Histological images with haematoxylin and eosin stain of control and infected mice were acquired by microscopy.

The animals were euthanized by applying deep intraperitoneal anaesthesia with xylazine and ketamine (10 mg/kg, 100 mg/kg) and later cervical dislocation. The whole heart was extracted and placed in a plastic container buffered with 10% formaldehyde and a pH of 7.2 for 24 h for proper fixation. Once the hearts were fixed, a coronal section was made to visualize the four cardiac chambers, the interventricular septum, and the outlets of the aorta and pulmonary arteries. Then, the samples were placed in a histological cassette for dehydration with alcohols at different concentrations. After, the hearts were placed in paraffin for cubes' preparation, and 5-μm-thick cuts were made with the aid of a microtome. These tissue sections were placed on a slide to be stained with haematoxylin and eosin, finally performing the assembly by placing a coverslip on the samples. Once the assembly of the cardiac tissue sections was completed, the samples were analysed using an optical microscope with a 40× objective, and photographs of the amastigote nests present were taken.

*2.2. Database—Histological Imaging*

In this work, the following image databases were processed: (a) image group for training and validation of the automatic segmentation algorithm for nests of amastigotes in mice inoculated with 1000 bloodstream trypomastigotes at 25, 30, and 35 days post-infection (Database I); (b) image groups for final testing consisting of three different groups under various conditions: one group of images consists of acquired images from uninfected control mice (Database II), another group of images was acquired from infected mice of an experimental model inoculated with 500 bloodstream trypomastigotes at 28, 29, and 30 days post-infection (Database III), and a final group of images was acquired from *T. cruzi*-infected mice inoculated with 500 bloodstream trypomastigotes at 31 days post-infection (Database IV), as shown in Table 1. All original RGB images have a resolution of 2592 × 1944 pixels.

**Table 1.** Description of the databases used for the training and validation stage of the automatic segmentation system for amastigote nests, as well as for final testing under various experimental conditions.

| Database | # of Mice | Inoculation Bloodstream Trypomastigotes | Post-Infection Days | # of Images | Infection Stage | Analysis |
|----------|-----------|------------------------------------------|---------------------|-------------|-----------------|----------|
| I | 9 | 1000 | 25/30/35 | 734 | Acute | Training & validation |
| II * | 3 | - | 0/120 | 27 | Acute | Final test 1 |
| III | 3 | 500 | 28/29/30 | 30 | Acute | Final test 2 |
| IV | 3 | 500 | 31 | 22 | Acute | Final test 3 |

* Healthy mice without amastigote nests present.

### 2.2.1. Database for Training and Validation

For the automatic segmentation of *T. cruzi* amastigote nests, a neural network was trained by means of a set of 734 histopathological microphotographs (Database I), which were taken from nine infected mice during the acute stage at 25, 30, and 35 days post-infection. Histopathological images were acquired to 40× from an Olympus CX23 microscope and a Leica DM750 microscope.

### 2.2.2. Databases for Final Tests

Three sets of images were considered to perform the final tests, where one set of images was acquired from healthy mice, while the other two sets of images belong to infected mice from different experimental models.

- **Database II.** This database consists of 27 images acquired from three healthy mice at 0 and 120 days after being administered physiological saline, showing no presence of amastigote nests in cardiac tissue. Therefore, they were considered as mice belonging to the control group. All histological images were acquired at 40× with a Leica DM750 microscope.
- **Database III.** For this final test of the nest segmentation system in the acute stage, images that were not previously seen by the algorithm during training were considered. For this group of images, three mice infected with 500 bloodstream trypomastigotes at 28, 29, and 30 days post-infection were considered. Thirty images of histological sections with presence of amastigote nests in cardiac tissue were processed. All histopathological images were acquired at 40× with a Leica DM750 microscope.
- **Database IV.** A final test was also considered using 22 microphotographs of histopathological sections from three infected mice with 500 bloodstream trypomastigotes at 31 days post-infection. Histopathological images were acquired at 40× with an Olympus CX23 microscope.

All the computational processing was conducted in the Laboratorio Universitario de Cómputo de Alto Rendimiento (LUCAR) and the Área de Análisis de Imágenes e Inteligencia Artificial of the Unidad Académica del IIMAS en Yucatán from the Universidad Nacional Autónoma de México (UNAM).

### 2.3. Binary Mask of the Manual Segmentation from Amastigote Nests

For each of the processed images, we manually generated their respective binary masks, segmenting each amastigote nest contained in the image; these annotations were considered the region of interest (ROI). In those images derived from histological slices of mice from the control group, i.e., healthy uninfected mice with no amastigote nests, images containing only background without any ROI were generated. The segmentation of regions used in the neural network training stage, to indicate to the neural network which patterns correspond to amastigote nests, was performed using MATLAB 2021a, while to perform final tests of the automatic segmentation and validation of spatial correspondence between the resulting automatic regions and the manual segmentation, masks were made with the help of ImageJ 1.53k software. All manually segmented ROIs, considered as ground truth masks, were reviewed and validated by experts in Chagasic histopathology.

### 2.4. Data Augmentation

One of the requirements for the robust training of a CNN is to provide the algorithm with a large number of input images [16]; however, it is not always possible to have a large database containing a robust number of images, especially when it comes to medical images. That is why techniques have been proposed for data augmentation through various types of transformations applied to the original images with which the neural network training is to be performed, thus generating a considerable increase in the number of input images. The advantages of making use of data augmentation are the model's generalization and improvement of the detection algorithm's precision, as well as its robustness for

unseen data. Some of the most common transformations applied to images are geometric ones, such as rotation or reflections, as well as contrast and brightness transformations or changes, which have been shown to give good results [17], in addition to spectral changes of the colour components, whether RBG, HSV, or any other colour model. For this work, data augmentation was performed from the training set of images, i.e., Database I (734 microphotographs of infected mice). Before performing the geometric, contrast, and colour transformations of the images, the original images were resized to reduce the computational weight of the segmentation algorithm, as were all the binary masks that were manually segmented. Subsequently, the vertical and horizontal axes, changes in contrast and brightness levels, and spectral changes in the RGB components were applied. Figure 2 shows a diagram of the data augmentation process that was carried out, generating a total database of 11,508 images for training and validation of the U-Net CNN. The following transformations were carried out during the data augmentation process: 90- and 180-degree rotations, random rotations between −45 and 45 degrees, reflections along the vertical and horizontal axes, changes in contrast and brightness levels, and spectral changes in the RGB components.

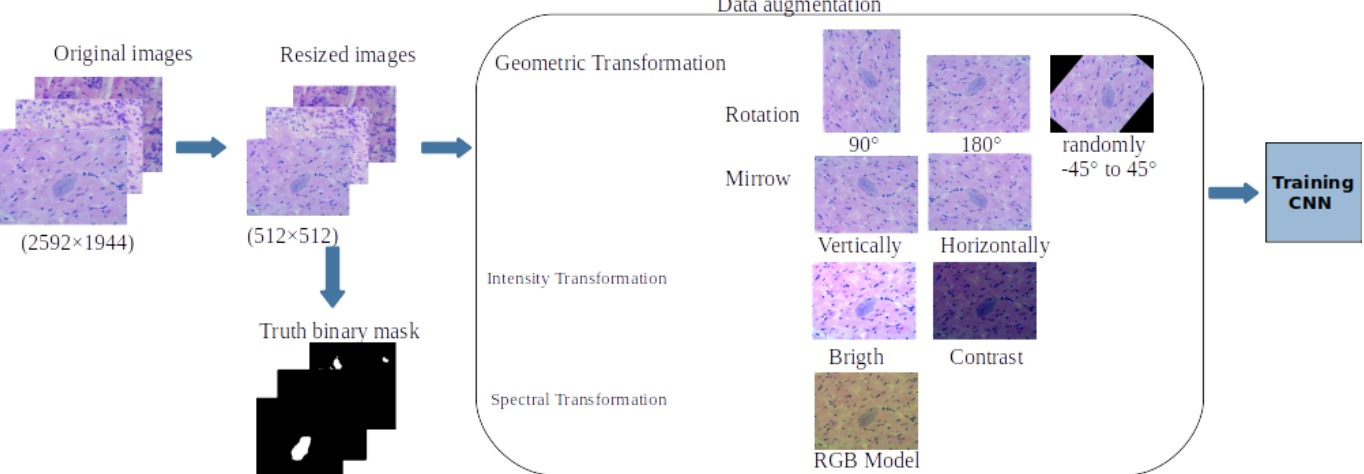

**Figure 2.** Diagram showing pre-processing of histological images and segmentation of binary masks, as well as representative examples of geometric, intensity, and spectral transformations to perform data enhancement.

*2.5. Deep Learning–Based Segmentation*

As part of what is known as deep learning (DL), neural networks are implemented based on basic elements such as interconnected perceptrons or artificial neurons performing functions as part of an artificial intelligence model. Artificial neural networks (ANNs) consist of an input layer, one or several hidden layers composed of a certain number of interconnected perceptrons, and an output layer, from which fully connected neural networks (FCNNs) can be obtained, where the pattern features are organized into vectors. As a result of this type of ANN model, new architectures have been considered for the analysis of images, taking into account the spatial relationship that exists between the pixels of an image, as well as existing patterns in it, such as corners, edges, or some other feature that allows differentiating between images. This is how a class of deep ANNs called convolutional neural networks (CNNs or ConvNets) arises. Currently, there are several proposed architectures whose input is an image, and whose output can be a data vector or another image [18]; for example, networks' architectures that propose dual-branch collaborative modules to extract global and local features of images [19], or spatial, spectral, and texture-aware attention networks from hyperspectral images information [20], but in the case of other applications.

### 2.5.1. U-Net Architecture

The U-Net architecture was presented by Ronneberger et al. [21]. Some characteristics of this CNN are to skip connections, concatenate feature maps, and add convolutions and non-linearities between the up-sampling steps. Skipped connections make it possible to recover the full spatial resolution in the network output, and it is for this reason that U-Net is a CNN architecture designed for region segmentation or detection. The U-Net architecture presents a symmetrical U-shaped structure and is divided into two paths: the encoder (contracting path), which provides classification information, and the decoder (expansive path), which allows the CNN to learn localized classification information. Figure 3 shows the U-Net architecture that was implemented for the amastigote nest segmentation algorithm. There are also concatenations or skipped connections between different layers from the encoder and the decoder, which convey feature maps from one to the other. In this work, the U-Net architecture was implemented because, in a preliminary work, it showed better performance compared to an SVM trained with texture features [22]. In addition, in other reported works the U-Net architecture has shown good results in the segmentation of histology image tissue [23,24].

In medical image analysis, the persistent problem of missing pixel-level context information can be solved by implementing the U-Net architecture [1,25]. As is shown in Figure 3, in the contraction path, first two $3 \times 3$ convolutions and a ReLU (rectified linear unit) activation function are applied, followed by a $2 \times 2$ max-pooling operation. This process is repeated five times, increasing the kernel size each time. And for the expansion process, the decoder consists of several up-samplings of the feature map using $2 \times 2$ up-convolution; after that, the feature map from the corresponding layer in the contracting path is concatenated onto the up-sampled feature map, followed by two successive $3 \times 3$ convolutions and ReLU activation functions, until the contraction path length is reached. At the end, an additional $1 \times 1$ convolution with two filters is used to reduce the feature map or feature vector to the desired number of classes to generate the output segmented image [21]. For this work, a total of 23 layers and 1,941,105 trainable U-Net parameters were implemented to obtain the segmentation maps ($512 \times 512$ pixels).

For the training of the U-Net, the input was the images of the histological microphotographs, each of which had its respective binary mask that indicated the spatial segmentation ground truth of the regions with the presence of amastigote nests.

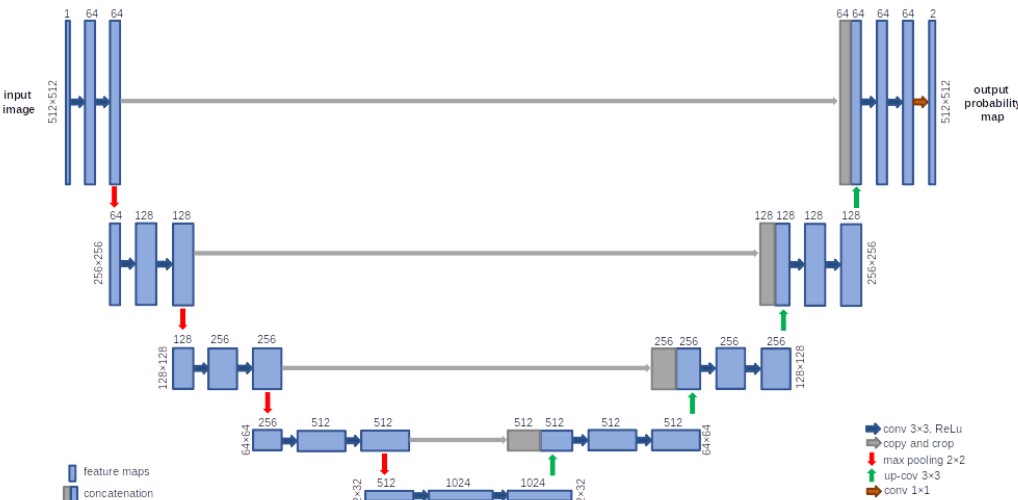

**Figure 3.** U-Net architecture, showing a symmetrical U-shaped structure, divided into the contracting path (encoder) and the expansive path (decoder), implemented for amastigote nest segmentation.

### 2.5.2. Selecting the Threshold for the Binary Segmentation Mask

As the output of the U-Net, what is obtained is an image corresponding to a probability map, where each pixel is assigned a probability that indicates how likely that pixel belongs

to an amastigote nest. Therefore, in order to have a binary mask of the segmentation of amastigote nests (Figure 4), it is necessary to consider an optimal threshold that allows us to binarize in the best way the resulting probability map at the output of the U-Net and thus obtain a binary segmentation of the amastigote nests in a robust way, for which it was proposed to carry out a precision-recall (PR) ROC curve analysis.

As post-processing, there is a thresholding stage of the probability map obtained as the output of the U-Net, which is why it is necessary to define the optimal threshold from which the best performance is obtained in the segmentation of the binary masks of the amastigote nests. For this purpose, precision-recall curves (PR curves) were constructed.

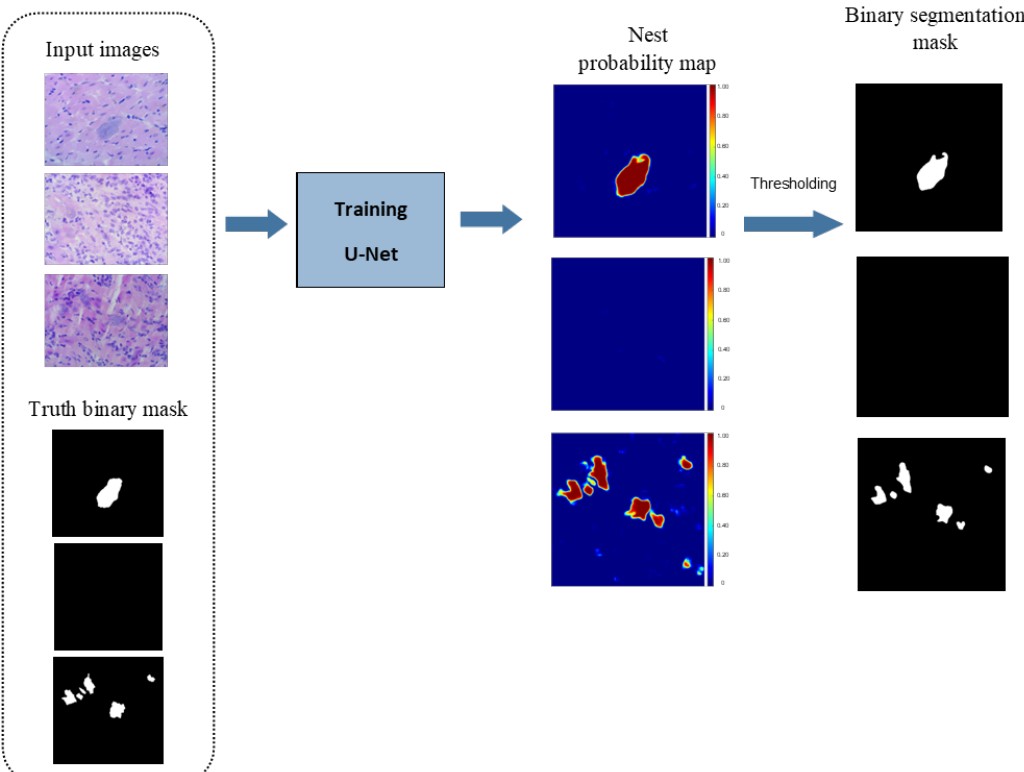

**Figure 4.** Diagram of the process of automatic segmentation of amastigote nests in histological microphotographs from a U-Net CNN architecture.

This is a parameterized curve that captures the trade-off between accuracy and presented noise, the threshold of which varies because precision (see Section 2.6.1) is the segmentation fraction that are true positives rather than false positives, while recall is the true positives fraction that were segmented rather than missed. The main advantage of using PR curves for the evaluation in segmentation overlap regions is that we can compare the produced segmentation with the same algorithm using different input parameters, where it has been proposed to vary the probability threshold ($t$) in order to be able to determine the threshold that is closest to a perfect model (precision = 1 and recall = 1) [26].

In addition, the values of the BF-Score and F1-score were verified (also described in Section 2.6.1), where the location of the maximum F-measure along a PR curve provides the optimal segmentation threshold given $t$, where this parameter determines the relative importance of each term ($t = 0.5$ indicates no preference for either) [27].

### 2.6. Training and Cross-Validation

To verify the reproducibility of the model and validate the results, a 5-fold cross-validation method was used [28], as is shown in Figure 5, where 90% of the images were used for training (n = 10,357) and 10% in the validation test (n = 1151). The parameters

chosen to train the U-Net network were an Adam gradient optimizer, a binary entropy loss function, and 50 epochs.

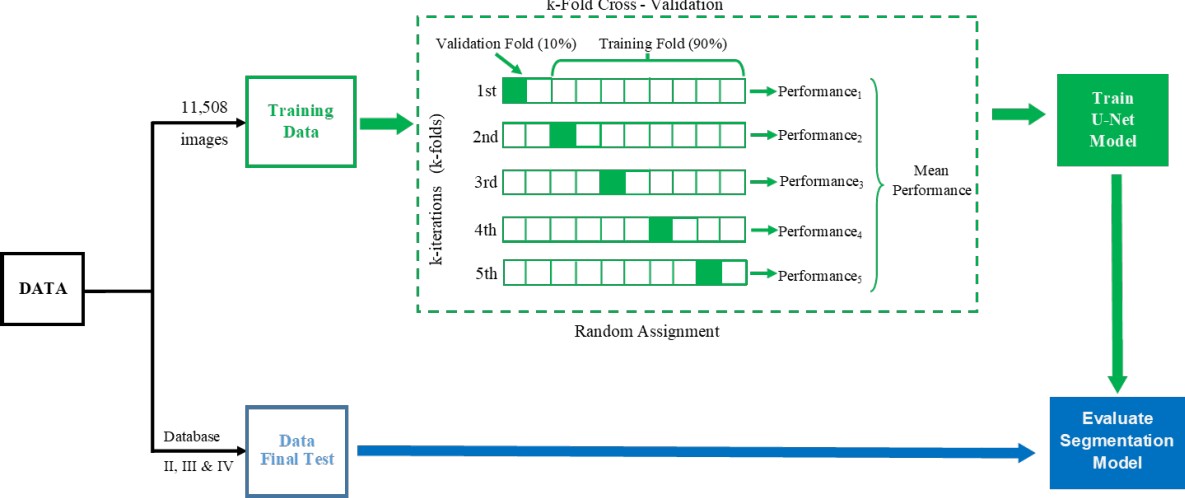

**Figure 5.** Diagram of training and 5-fold cross-validation for automatic amastigote nest segmentation; the final test used three different databases.

2.6.1. Confusion Matrix Metrics and Similarity Segmentation Indices

A classifier's performance can be summarized by means of a confusion matrix, where each row refers to ground truth classes as recorded in the reference set, and each column to classes as predicted by the classifier. It is convenient to distinguish the performance of the classes, where correctly classified positives and negatives are referred to as true positives (TP) and true negatives (TN), respectively; incorrectly classified positives are called false negatives (FN); and misclassified negatives are called false positives (FP) [29], as is shown in Table 2.

**Table 2.** Confusion matrix representation.

|  |  | Predicted Nest | |
|---|---|---|---|
|  |  | 1 | 0 |
| **Truth** | 1 | True Positive (TP) | False Negative (FN) |
| **Nest** | 0 | False Positive (FP) | True Negative (TN) |

Some performance measures to validate the ROI segmentation can be defined based on the confusion matrix as follows:

$$\text{Accuracy} = \frac{\text{TP} + \text{TN}}{\text{TP} + \text{TN} + \text{FP} + \text{FN}}, \tag{1}$$

$$\text{Precision} = \frac{\text{TP}}{\text{TP} + \text{FP}}, \tag{2}$$

$$\text{Sensitivity} = \frac{\text{TN}}{\text{FP} + \text{TN}}, \tag{3}$$

$$\text{Recall} = \frac{\text{TP}}{\text{TP} + \text{FN}}. \tag{4}$$

Also, to be able to compare the similarity between the ground truth mask of segmented regions and the mask automatically predicted by U-Net's CNN of amastigote nests, statistical similarity coefficients can be obtained, such as the Dice and Jaccard in-

dices [30,31], which, in this case, are measures of the relationship between areas of the manually segmented and predicted binary masks:

$$\text{Dice} = \frac{2TP}{2TP + FP + FN}, \tag{5}$$

$$\text{Jaccard} = \frac{TP}{TP + FP + FN}. \tag{6}$$

The accuracy and Jaccard coefficient were implemented for the training stage evaluating Database I, and all similarity measures were used for the final tests using Databases II, III, and IV, as is explained in the next section. As mentioned in Section 2.5.2, for the case of the BF-score and F1-score values, the location of the maximum F-measure along the PR curve provides the optimal segmentation threshold given t, where the difference between the BF-score and F1-score is that the first score is a boundary-based measure, while the F1-score is an F-measure that considers all the pixels in a region and is equivalent to the Dice similarity coefficient [32]; these scores are valued between 0 and 1, where larger values are more desirable, and are defined as:

$$\text{F1-score} = \frac{2TP}{2TP + FP + FN}, \tag{7}$$

$$\text{BF-score} = \frac{(2\text{Precision})(\text{Recall})}{\text{Precision} + \text{Recall}}. \tag{8}$$

*2.7. Final Test Validation*

As can be seen in Figure 5, in a second stage after having performed the training of the U-Net and the cross-validation to know its performance, some final tests of the implemented algorithm were performed using images that were not seen by the algorithm during the training stage. These are new images from which the U-Net's performance for automatic segmentation of amastigote nests is objectively analysed. For this purpose, we performed this final test using three different sets of images: histological images of healthy mice belonging to the control group (Database II) and histopathological images of infected mice from two different experimental models (Databases III and IV). All images were pre-processed by means of a scaling transformation to reduce their original size ($512 \times 512$ pixels), with the intention that they would have the required size for the implemented U-Net and could be used as input for this CNN.

From the image resulting from the automatic segmentation by the U-Net, the binary mask of the region or regions corresponding to the parasite amastigote nests was obtained; then, these binary masks were validated with the ground truth images to obtain as performance metrics the accuracy, precision, sensitivity, and recall, as well as the Dice and Jaccard similarity indices described above.

**3. Results and Discussion**

The U-Net performance for amastigote nest segmentation in the training stage for the 5-fold cross-validation showed an accuracy of $98.19 \pm 0.01\%$ and a Jaccard coefficient value of $49.43 \pm 0.00\%$. From these results, we can observe that during the training stage, for almost all validation images, the U-Net was capable of detecting the amastigote nests well, but not the total number of pixels that contained the segmented nest regions. Figure 6 shows two cases of nest segmentation during the training stage, where we can observe one example of successful segmentation and another example of poor segmentation, which produced a bad overlap between the binary masks of ground truth segmentation and the automatic segmentation of nest regions.

From Figure 6, it can be seen that in one of the examples, the number of pixels considered as part of a parasite nest was overestimated; however, it is also possible to observe that some of them belong to possible very small nests that are found in the

histological image but were not considered by the expert, precisely because of their size, when performing the manual segmentation and the subjectivity that some of these regions presented. Likewise, it was observed that the cases that presented low Jaccard coefficient values were in histological images that presented some artefacts or very particular RGB colour spectral patterns, derived from the staining quality of the histological sample rather than the image acquisition or processing.

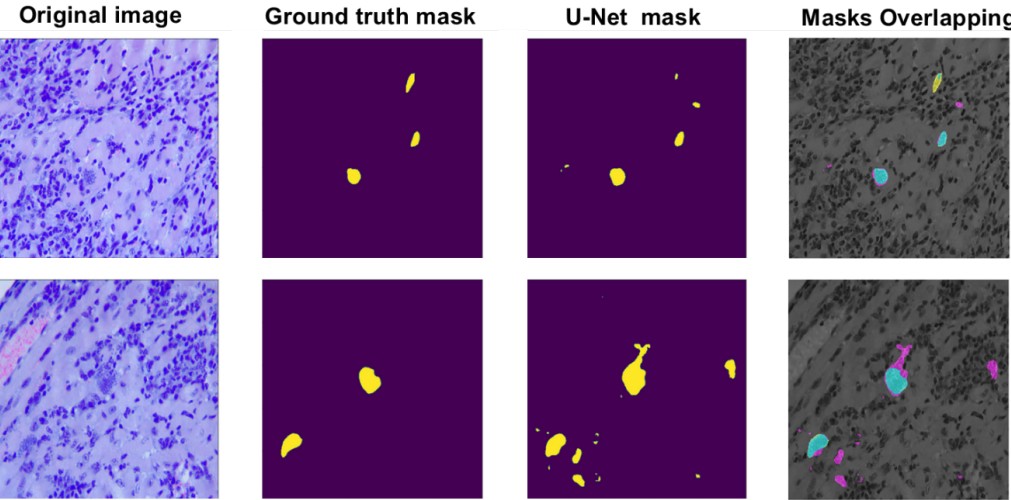

**Figure 6.** Two examples of amastigote nest segmentation by a U-Net architecture during the training validation: an example of very successful segmentation (**upper** row), and an example of over-segmentation (**lower** row). For the visualization of mask overlapping: the ground truth region is shown in yellow, the automatic region in rose, and the overlapping region in cyan.

However, these results may vary according to the probability threshold implemented in the probability map at the output of the U-Net, with the objective to obtain the binary mask corresponding to the segmented region of the amastigote nest. Since it is necessary to analyse which is the optimal threshold to achieve the best learning system performance for the amastigote nests' segmentation, PR-curves were generated considering 52 histopathological images of infected mice (Database III + Database IV), from which the corresponding binary masks were generated using different probability values as thresholds ($t$ = 0.97, 0.90, 0.85, 0.8 0, 0.75, 0.70, 0.65, 0.60, 0.50, 0.40, 0.30, 0.20, 0).

The automatic segmentation system's performance was analysed to obtain the precision and recall for each threshold $t$. The final PR-curve is shown in Figure 7a, showing the optimal threshold $t$ = 0.20 that allows us to have better performance in detecting the nests. Figure 7b shows the graph of the BF-score and F1-score values, corresponding to each of the used $t$ threshold values, observing that the highest scores are obtained for a value of $t$ = 0.20. We can also observe that the probability map obtained by a U-Net architecture is quite robust, since it is possible to detect a nest from low probability values, that is, practically a zero value in the probability map corresponds to regions that are not amastigote nests.

Table 3 shows the performance metrics of the U-Net for the generation of the binary masks of the amastigote nests' segmentation, as well as the similarity coefficients, which were obtained when performing the final tests using Databases II, III, and IV considering a threshold value of $t$ = 0.20 in the probability map.

**Table 3.** Confusion matrix metrics and similarity coefficients for final tests using Databases II, III, and IV with the implemented U-Net architecture (*t* = 0.20).

| Database | Accuracy | Precision | Specificity | Recall | Dice | Jaccard |
|---|---|---|---|---|---|---|
| II * | $0.999 \pm 0.001$ | - | $1.00 \pm 0.00$ | - | - | - |
| III | $0.988 \pm 0.009$ | $0.66 \pm 0.14$ | $0.96 \pm 0.14$ | $0.75 \pm 0.20$ | $0.69 \pm 0.15$ | $0.55 \pm 0.16$ |
| IV | $0.991 \pm 0.008$ | $0.70 \pm 0.13$ | $0.98 \pm 0.01$ | $0.94 \pm 0.07$ | $0.80 \pm 0.09$ | $0.67 \pm 0.12$ |

\* Healthy mice without amastigote nests present.

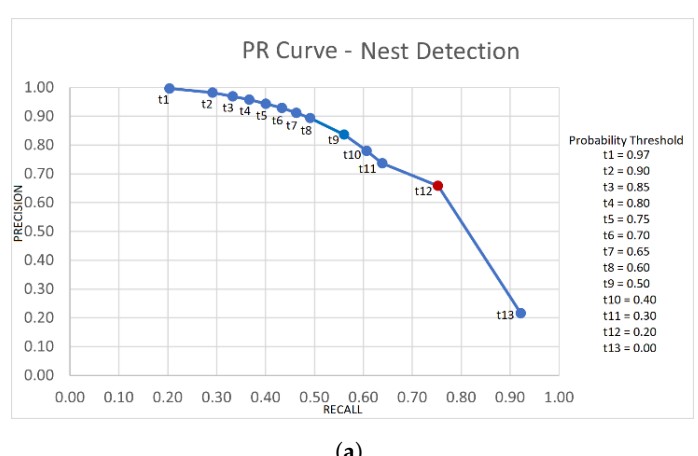

(**a**)

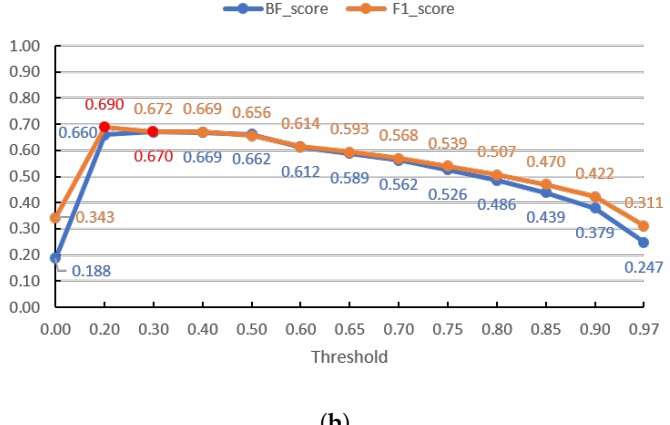

(**b**)

**Figure 7.** Performance graphs of the automatic amastigote nest segmentation for different thresholding values *t* from the probability map in the case of the general infected group (Databases III + IV): (**a**) PR curve, (**b**) BF- and F1-scores; where in each graph the points and values corresponding to the optimal threshold are marked in red.

As can be seen in Table 3, in the case of images corresponding to healthy mice (Database II) that did not present amastigote nests, in most cases, no binary regions or masks were detected, but only the background; with the exception of two images in which a segmented region was obtained, as shown in Figure 8.

We associate this erroneous detection with the RGB colour pattern present in the images derived from the staining process of the histological samples; those small regions present similar characteristics to those of an amastigote nest. Therefore, as future work, more extensive RGB colour transformations and corrections can be carried out for the U-Net training. In Figure 9, we can see that in relation to the other two final tests carried out with infected mice from two different *T. cruzi* experimental infection models, segmentation of all regions belonging to amastigote nests was achieved; however, in some cases, a subadjustment of the segmented regions was presented, or regions related to small nests that had not been considered as such were detected. This could make it possible to find regions

associated with amastigote nests' presence at an early infection stage, which would result in an auxiliary tool for the clinician.

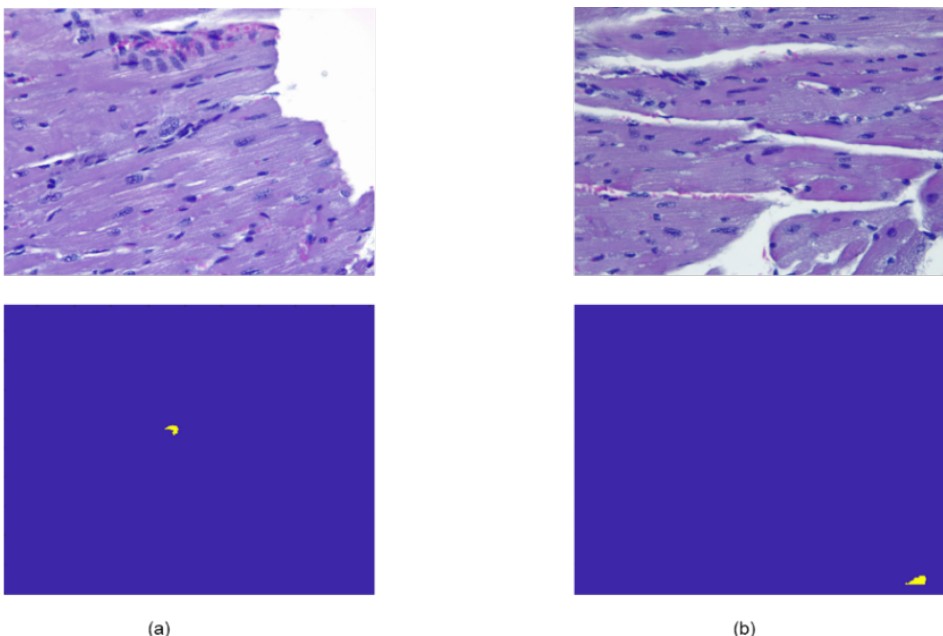

(a)

(b)

**Figure 8.** Examples of the two cases of histological images from the control mice group, without amastigote nests present, where the U-Net architecture segmented some pixels as a nest. Subfigure (**a**) shows the original image of a healthy mouse and the segmentation with false positives detected; subfigure (**b**) is another representative example of misclassified data in a control mouse image.

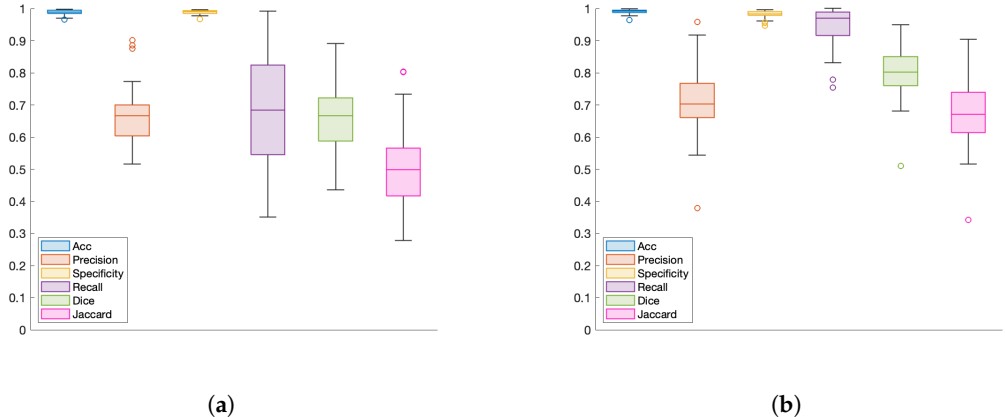

(**a**)

(**b**)

**Figure 9.** U-Net performance metrics for automatic nest segmentation: (**a**) Database III (acute stage, 500 bloodstream trypomastigotes, 28, 29, and 30 days post-infection), and (**b**) Database IV (acute stage, 500 bloodstream trypomastigotes, 31 days post-infection).

The detail of the average pixels correctly classified for Databases III and IV is shown in Table 4. The results of Database II are not presented since it is a healthy population and does not contain amastigote nests. Similar to what is shown in Table 3 and Figure 9, a high rate of correctly classified pixels can be observed (98.8% for Database III and 99.1% for Database IV adding TP and TN data). False negatives (0.64% for Database III and 0.12% for dataset IV) are associated with small incorrect detections of altered color patterns derived from the staining process in histology images. False positives can also be observed (0.60% and 0.82% for Databases III and IV respectively); in some cases the FPs are due to misclassified data, and in others they are regions that were not manually annotated by the experts, but are detected by the network. The obtained results suggest that the proposed

methodology can be useful as a tool for the experimental study of Chagas disease, as well as to quantify the degree of amastigote nests in different regions of the heart.

Another point to highlight is the good performance after resizing the images. This resizing was done in order to mitigate the computational cost, as well as the transmission and storage of information. In addition, this normalization allows the processing of images acquired with microscopes of different centers, which may have different resolutions. It is considered that these factors can facilitate the use of this tool in different Chagas disease research centers.

**Table 4.** Confusion matrices obtained from the final tests with Databases III and IV respectively. Data belonging to a nest are represented as 1 and healthy tissue as 0. Results are shown according to the number of pixels classified (mean ± standard deviation).

| | | Predicted Nest Database III | | Predicted Nest Database IV | |
|---|---|---|---|---|---|
| | | 1 | 0 | 1 | 0 |
| **Truth Nest** | 1 | 3104 ± 1933 | 1668 ± 1514 | 5161 ± 3556 | 306 ± 352 |
| | 0 | 1573 ± 1058 | 255796 ± 4034 | 2140 ± 2022 | 254536 ± 4922 |

Since no other work has been found that presents a method for amastigote nests' segmentation for *T. cruzi* infection, we do not report any direct and objective comparison of the results.

However, in accordance with the state of the art, the importance of using computational techniques in similar applications can be emphasized, such as a work related to the semi-automatic counting of amastigote nests by mathematical morphology; although in other studies the authors do not perform nest detection, they show a precision of 0.91 and 0.96, an accuracy of 0.83 and 0.86, as well as accuracy rates of 0.85 ± 0.10, recall rates of 0.86 ± 0.11%, and error rates of 0.16 ± 0.08 [33]. A similar work has also been presented about amastigote counting in Chagas-infected cells using unsupervised automatic classification techniques and morphological granulometric processing, obtaining in its test models error rates of 0.10 and 0.26, precision of 0.76 and 0.85, recall 0.61 and 0.78, as well as F-measures of 0.66 and 0.75, respectively [34]. Therefore, we consider that our results are objectively comparable to those already reported, also considering that automatic nest detection is being carried out.

Regarding the use of the U-Net architecture, similar works have been reported with satisfactory results, specifically in the case of *T. cruzi* parasite segmentation by an automatic system to detect the parasite in blood sample images, but not in the parasite stage as an amastigote in cardiac tissue to relate it to heart damage; even so, good results are reported in the use of the U-Net (F2 = 0.80, recall = 0.87, precision = 0.63, and Dice = 0.68) [13]. It is more common to find works related to automatic *T. cruzi* parasite detection in blood [35,36]; however, this analysis is far from the objective of our work.

It has also been found that there are some works related to the *T. cruzi* parasite, but making use of fluorescence images, where the characteristics or observed patterns differ from those present in a histological microphotograph, where image processing techniques have been used for image enhancement by intensity equalization and shading correction to calculate the ratio between infected host cells and the total host cells, and the number of parasites per infected host cell [37].

There are different clinical scenarios in which histopathology for *T. cruzi* infection diagnosis in tissues is very useful: heart transplantation is a therapeutic option in patients with CD where there is advanced heart failure that is refractory to medical therapy, and monitoring in transplant patients, since infection reactivation can occur. In addition, infection reactivation can occur under immunosuppressive conditions, as is the case of coinfections with Chagas/HIV, where in approximately 30–40% of the cases, the infection reactivation involves the heart. Other immunosuppressive pathologies that can reactivate the infection are autoimmune diseases, cancer and cancer chemotherapy, immunosuppres-

sive drugs administered to prevent rejection in the case of transplants, as well as any other immunosuppressive disease. In these cases, the pathological findings described are acute myocarditis, intense inflammatory infiltrate, damage to cardiac fibres, focal necrosis, and a large number of amastigote nests, which may be the joint result of established infection and acute infection reactivation [38].

The identification of amastigote nests by histopathology, from tissues of biopsies or cadavers, is one of the most used diagnostic tools, since not only the number of amastigote nests and their distribution can be identified, but also the inflammation extension and repair, thus establishing indices where the survival of affected patients can be predicted. However, according to our experience in the laboratory, we have observed that for robust amastigote nest identification in necrotic and inflamed cardiac tissues, a high degree of preparation is required, since *T. cruzi* amastigote nests can commonly be confused with other microorganisms that have a similar shape and size, such as *Histoplasma capsulatum*, *Toxoplasma gondii*, or *Leishmania donovani* amastigotes [39]. Therefore, using this deep U-Net network has been of great help to identify and calculate, in a robust and fast way, the number and distribution of amastigote nests in the evaluated tissues.

## 4. Conclusions

In this paper, we present an implementation of the U-Net deep neural network, in combination with data augmentation approaches, for the automatic segmentation of amastigote nests in histopathology images. Adequate nest segmentation can help experts with CD study, follow-up, and analysis. For the automatic segmentation evaluation, four databases were used: the first was used for training, and the other three databases to carry out the final validation with data not previously seen by the system. These three image sets used for the final test contain images with and without the parasite nests present (infected and control groups, respectively), which were acquired under different conditions. This exhaustive validation has helped to evaluate the proposed approach, as well as its robustness to images with different histology stains and conditions, showing good results. The U-Net convolutional network implementation has made it possible to successfully segment *T. cruzi* amastigote nests in the acute stage using histopathological images of cardiac tissue in a murine experimental model. The automatic segmentation from this deep learning technique has presented very favourable results and can be used as an auxiliary tool in the study, analysis, and diagnosis of subjects infected with the parasite, seeking to carry out this analysis in an early infection stage. Thus, this automatic detection and segmentation can be used not only in experimental studies but also directly in patients suffering from CD, particularly in those in which an infection reactivation is studied. The results shown, in some cases, present an overestimation of amastigote nests that were not initially labelled by the experts due to their small size; hence, we believe this had an impact on the Dice and Jaccard indices' performance. However, the segmentation of these amastigote nests without manual labelling confirms the importance of using automatic algorithms for segmentation and exploration of histopathological images. On the other hand, the final test with the control image database (Database II without the presence of nests) helped to verify that the automatic system implemented did show a low rate of false positives, as was observed in the results obtained.

As future work, it is proposed to evaluate other neural network architectures such as Mask-RCNNN, residual neural networks, based on transformers or self-trained networks. In addition, it is expected to further evaluate the presented system with image banks from other centers and/or images with other characteristics (different resolutions, staining, acquired at different magnifications of the microscope, or quantity of inoculated trypomastigotes). This would allow the use of the tool in different centers.

In conclusion, it is considered that the proposed automatic segmentation tool can be useful for the detection and quantification of amastigote nests, which could help the study of Chagas disease in different clinical centers.

**Author Contributions:** Conceptualization, N.H.-M. and P.H.; Data curation, N.H.-M. and L.G.-C.; Formal analysis, N.H.-M. and J.P.-G.; Funding acquisition, N.H.-M. and J.P.-G.; Investigation, P.H.; Methodology, N.H.-M. and J.P.-G.; Project administration, N.H.-M.; Resources, N.H.-M., P.H. and L.G.-C.; Software, N.H.-M. and J.P.-G.; Supervision, N.H.-M.; Validation, N.H.-M. and L.G.-C.; Visualization, J.P.-G.; Writing—original draft, N.H.-M.; Writing—review & editing, P.H., L.G.-C. and J.P.-G. All authors have read and agreed to the published version of the manuscript.

**Funding:** This work was supported by UNAM-PAPIIT Programs IT101422 and IA104622. The APC was funded by IT101422.

**Institutional Review Board Statement:** The experiment was approved by the ethical committee of the Centro de Investigaciones Regionales—Universidad Autónoma de Yucatán (CIRB-006-2017) (CEI-04-2020). The animals were handled according to the Guide for the Care and Use of Laboratory Animals (eighth edition) National academies 2011.

**Informed Consent Statement:** Not applicable.

**Data Availability Statement:** The data used in this research can be shared via email.

**Acknowledgments:** The authors thank the Parasitology and Zoonoses Laboratories Staff of the Centro de Investigaciones Regionales Hideyo Noguchi at the Universidad Autonoma de Yucatan for help with data acquisition; N. Sanchez-Patiño, A. Toriz-Vazquez, E. J. Rosado-Sanchez, and M. Burquez for technical support, and to acknowledge technical assistance of Adrian Duran Chavesti for LUCAR server management.

**Conflicts of Interest:** The authors declare no conflict of interest.

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
