# Peer review of "Deep Learning–Based Segmentation of Trypanosoma cruzi Nests in Histopathological Images"

_electronics, doi:10.3390/electronics12194144_

Round 1
Reviewer 1 Report
In this article, the authors apply a Deep Learning algorithm (U-Net) to segment histopathological images. The results obtained demonstrate the relevance of this choice.
However, some questions remain:
1) Various image segmentation algorithms based on Deep-Learning approaches exist, such as Mask-RCNN. Also, what is the superiority of U-Net over Mask-RCNN in the context of the application described in this article?
2) The original 2592x1944 pixel images are resized to 512x512 pixels. Doesn't this result in a loss of precision? Wouldn't it be better to keep the original resolution and split the original image into several 512x512 pixel regions of interest?
3) Confusion matrices should be added to the results, and the discussion should be completed taking these matrices into account.
Author Response
Manuscript ID: electronics-2629353
Type of manuscript: ArticleTitle: Deep Learning–Based Segmentation of Trypanosoma cruzi Nests in Histopathological Images
Authors: Nidiyare Hevia-Montiel *, Paulina Haro, Leonardo Guillermo-Cordero, Jorge Perez-Gonzalez
In this cover letter we explain and respond, point by point, to the referees’ comments to improve the manuscript. We appreciate the comments and suggestions from the reviewers.
List of the modifications
Reviewer 1
In this article, the authors apply a Deep Learning algorithm (U-Net) to segment histopathological images. The results obtained demonstrate the relevance of this choice.
However, some questions remain:
Comment 1.1
Various image segmentation algorithms based on Deep-Learning approaches exist, such as Mask-RCNN. Also, what is the superiority of U-Net over Mask-RCNN in the context of the application described in this article?
Authors answer 1.1
Thanks to the reviewer for this comment. A U-Net architecture was selected in this work because it is a network specifically designed for biomedical image segmentation [1]. Moreover, in a preliminary work the U-Net architecture showed better performance than other machine learning based algorithms such as a support vector machine trained with texture features [2]. The Mask-RCNN architecture is a general purpose network that combines Faster R-CNN and a Fully Convolutional Network (FCN), is data intensive and due to the combination of two networks has a high computational cost [3]. Moreover, in several works in which different tissues are segmented in histology images, the U-Net architecture showed equal or better performances than Mask-RCNN [4-5]. However, as shown in [5], a combination of both architectures can obtain better performances. Therefore, as future work it is proposed to implement new architectures such as Mask-RCNN, or networks based on residual elements or transformers for example.
To address this comment in the manuscript, an explanation has been added in section 2.5.1 and in the conclusions section.
[1] Liu, X.; Song, L.; Liu, S.; Zhang, Y. A Review of Deep-Learning-Based Medical Image Segmentation Methods. Sustainability 2021, 13, 1224. https://doi.org/10.3390/su13031224
[2] N. Sanchez-Patiño, A. Toriz-Vazquez, N. Hevia-Montiel and J. Perez-Gonzalez, "Convolutional Neural Networks for Chagas’ Parasite Detection in Histopathological Images," 2021 43rd Annual International Conference of the IEEE Engineering in Medicine & Biology Society (EMBC), Mexico, 2021, pp. 2732-2735, doi: 10.1109/EMBC46164.2021.9629563.
[3] K. He, G. Gkioxari, P. Dollár and R. Girshick, "Mask R-CNN," 2017 IEEE International Conference on Computer Vision (ICCV), Venice, Italy, 2017, pp. 2980-2988, doi: 10.1109/ICCV.2017.322.
[4] Putzu, L.; Fumera, G. An Empirical Evaluation of Nuclei Segmentation from H&E Images in a Real Application Scenario. Appl. Sci. 2020, 10, 7982. https://doi.org/10.3390/app10227982
[5] A. O. Vuola, S. U. Akram and J. Kannala, "Mask-RCNN and U-Net Ensembled for Nuclei Segmentation," 2019 IEEE 16th International Symposium on Biomedical Imaging (ISBI 2019), Venice, Italy, 2019, pp. 208-212, doi: 10.1109/ISBI.2019.8759574.
Comment 1.2
The original 2592x1944 pixel images are resized to 512x512 pixels. Doesn't this result in a loss of precision? Wouldn't it be better to keep the original resolution and split the original image into several 512x512 pixel regions of interest?
Authors answer 1.2
We appreciate this important comment. It is expected that the proposed tool for the segmentation of amastigotes nests will be used in several Chagas disease research centers (for this reason the system was validated with multiple databases with different properties). It is considered that each center may have different acquisition equipment with different image resolution qualities. Therefore, for the proposed segmentation tool, it was considered relevant to "normalize" the images to a lower resolution, and thus ensure good performance with image sets of different characteristics. In addition, the clinical research centers where the tool would be used do not have efficient computer equipment in terms of performance, transmission and storage of information. For that reason, compressing the images helps to mitigate these drawbacks. However, this is a good point that should be considered in future developments. A detailed explanation of image normalization/compression has been added to the manuscript.
Comment 1.3
Confusion matrices should be added to the results, and the discussion should be completed taking these matrices into account.
Authors answer 1.3
Thanks for the suggestion. The confusion matrices of databases III and IV (unseen external data) have been added to the results section (Table 4). In addition, the respective discussion of these results has been added.
Reviewer 2 Report
In this paper, a method that combines U-Net deep neural network with data augmentation is proposed to automatically segment amastigote nests in histopathology images. The method was trained on histological images from an acute-stage murine experimental model using a 5-fold cross-validation approach and was subsequently tested on new data. The experimental results demonstrate that the U-Net architecture can be effective for histopathology image diagnosis and analysis. However, several issues have been identified:
1. The author is requested to summarize the innovative points into three points in the "Introduction" section.
2. The methods introduced in "Introduction" of this paper are not novel enough, some latest methods should be added. (For details, please refer to and quote the following " Accurate segmentation of nuclei in pathological images via sparse reconstruction and deep convolutional networks "," Dual-branch collaborative learning network for crop disease identification " and " Cost-sensitive multi-task learning for nuclear segmentation and classification with imbalanced annotations ")
3. The font size in Figure 2 is small, please optimize Figure 2 to achieve a more intuitive effect. ( please refer to " Spatial, Spectral, and Texture Aware Attention Network Using Hyperspectral Image for Corn Variety Identification ")
4. Please mark what (a) and (b) in Figure 8 represent respectively.
5. The citation format in the section " 2.5.1. U-Net architecture" should correspond to the labeling of the figure. If the Figure is labeled "Figure 3," the text should also be labeled "Figure 3" instead of "Figure ??". Please check the other parts carefully for similar errors and make corrections.
6. Every formula in the article should be punctuated with a ", "or ".". Please check and proofread all formulas carefully.
7. When multiple letters are used together as variables in an equation, they should be in regular (non-italic) font. The author should review carefully and revise all equations in the article to ensure consistent formatting.
8. Most of the references cited in this paper are old and fail to reflect the cutting-edge of the research. Please update and supplement the references.
In this paper, a method that combines U-Net deep neural network with data augmentation is proposed to automatically segment amastigote nests in histopathology images. The method was trained on histological images from an acute-stage murine experimental model using a 5-fold cross-validation approach and was subsequently tested on new data. The experimental results demonstrate that the U-Net architecture can be effective for histopathology image diagnosis and analysis. However, several issues have been identified:
1. The author is requested to summarize the innovative points into three points in the "Introduction" section.
2. The methods introduced in "Introduction" of this paper are not novel enough, some latest methods should be added. (For details, please refer to and quote the following " Accurate segmentation of nuclei in pathological images via sparse reconstruction and deep convolutional networks "," Dual-branch collaborative learning network for crop disease identification " and " Cost-sensitive multi-task learning for nuclear segmentation and classification with imbalanced annotations ")
3. The font size in Figure 2 is small, please optimize Figure 2 to achieve a more intuitive effect. ( please refer to " Spatial, Spectral, and Texture Aware Attention Network Using Hyperspectral Image for Corn Variety Identification ")
4. Please mark what (a) and (b) in Figure 8 represent respectively.
5. The citation format in the section " 2.5.1. U-Net architecture" should correspond to the labeling of the figure. If the Figure is labeled "Figure 3," the text should also be labeled "Figure 3" instead of "Figure ??". Please check the other parts carefully for similar errors and make corrections.
6. Every formula in the article should be punctuated with a ", "or ".". Please check and proofread all formulas carefully.
7. When multiple letters are used together as variables in an equation, they should be in regular (non-italic) font. The author should review carefully and revise all equations in the article to ensure consistent formatting.
8. Most of the references cited in this paper are old and fail to reflect the cutting-edge of the research. Please update and supplement the references.
Author Response
Manuscript ID: electronics-2629353
Type of manuscript: ArticleTitle: Deep Learning–Based Segmentation of Trypanosoma cruzi Nests in Histopathological Images
Authors: Nidiyare Hevia-Montiel *, Paulina Haro, Leonardo Guillermo-Cordero, Jorge Perez-Gonzalez
In this cover letter we explain and respond, point by point, to the referees’ comments to improve the manuscript. We appreciate the comments and suggestions from the reviewers.
List of the modifications
Reviewer 2
In this paper, a method that combines U-Net deep neural network with data augmentation is proposed to automatically segment amastigote nests in histopathology images. The method was trained on histological images from an acute-stage murine experimental model using a 5-fold cross-validation approach and was subsequently tested on new data. The experimental results demonstrate that the U-Net architecture can be effective for histopathology image diagnosis and analysis. However, several issues have been identified:
Comment 2.1
The author is requested to summarize the innovative points into three points in the "Introduction" section.
Authors answer 2.1
Thanks for this recommendation. The most innovative points of this research have been summarized in the introduction section.
Comment 2.2
The methods introduced in "Introduction" of this paper are not novel enough, some latest methods should be added. (For details, please refer to and quote the following " Accurate segmentation of nuclei in pathological images via sparse reconstruction and deep convolutional networks "," Dual-branch collaborative learning network for crop disease identification " and " Cost-sensitive multi-task learning for nuclear segmentation and classification with imbalanced annotations ")
Authors answer 2.2
Thanks for this recommendation. The introduction section was completed adding some latest methods, as you recommended :
“Histopathological images are an excellent use case for application of deep learning strategies, where a first challenge has been to analyze individual cells for accurate diagnosis from deep convolutional neural network to robustly and accurately detect and segment cells, implementing cascaded by multi-layer convolution operation without subsampling layers for cell segmentation [1]; or some cell segmentation works that tackle the data heterogeneity problem by cost-sensitive learning strategy to solve the imbalanced data distribution, and post-processing step based on the controlled watershed to alleviate fragile cell segmentation with unclear contour [2].”
[ 1] Xipeng Pan, Lingqiao Li, Huihua Yang, Zhenbing Liu, Jinxin Yang, Lingling Zhao, Yongxian Fan, Accurate segmentation of nuclei in pathological images via sparse reconstruction and deep convolutional networks, Neurocomputing, Volume 229, 2017, Pages 88-99, ISSN 0925-2312, https://doi.org/10.1016/j.neucom.2016.08.103.
[2] Xipeng Pan, Jijun Cheng, Feihu Hou, Rushi Lan, Cheng Lu, Lingqiao Li, Zhengyun Feng, Huadeng Wang, Changhong Liang, Zhenbing Liu, Xin Chen, Chu Han, Zaiyi Liu, SMILE: Cost-sensitive multi-task learning for nuclear segmentation and classification with imbalanced annotations, Medical Image Analysis, Volume 88, 2023, 102867, ISSN 1361-8415, https://doi.org/10.1016/j.media.2023.102867.
In 2.5 section was included the next paragraph and references:
“For example, networks' architectures that propose dual-branch collaborative modules to extract global and local features of images [3], or spatial, spectral, and texture-aware attention networks from hyperspectral images information [4], but in the case of other applications.”
[3] Zhang W, Sun X, Zhou L, Xie X, Zhao W, Liang Z, Zhuang P. Dual-branch collaborative learning network for crop disease identification. Front Plant Sci. 2023 Feb 10;14:1117478. doi: 10.3389/fpls.2023.1117478. PMID: 36844059; PMCID: PMC9950499
[4] W. Zhang, Z. Li, H. -H. Sun, Q. Zhang, P. Zhuang and C. Li, "SSTNet: Spatial, Spectral, and Texture Aware Attention Network Using Hyperspectral Image for Corn Variety Identification," in IEEE Geoscience and Remote Sensing Letters, vol. 19, pp. 1-5, 2022, Art no. 5514205, doi: 10.1109/LGRS.2022.3225215.
Comment 2.3
The font size in Figure 2 is small, please optimize Figure 2 to achieve a more intuitive effect. (please refer to " Spatial, Spectral, and Texture Aware Attention Network Using Hyperspectral Image for Corn Variety Identification ")
Authors answer 2.3
The font size in Figure 2 has been optimized and improved to be more intuitive, and the reference was included in section 2.5.
Comment 2.4
Please mark what (a) and (b) in Figure 8 represent respectively.
Authors answer 2.4
The detail of subfigures a and b has been added in the caption of Figure 8.
Comment 2.5
The citation format in the section " 2.5.1. U-Net architecture" should correspond to the labeling of the figure. If the Figure is labeled "Figure 3," the text should also be labeled "Figure 3" instead of "Figure ??". Please check the other parts carefully for similar errors and make corrections.
Authors nswer 2.5
Section citation formatting, as well as figure and table labels have been revised and corrected throughout the document.
Comment 2.6
Every formula in the article should be punctuated with a ", "or ".". Please check and proofread all formulas carefully.
Authors answer 2.6
The format of all equations has been reviewed and corrected.
Comment 2.7
When multiple letters are used together as variables in an equation, they should be in regular (non-italic) font. The author should review carefully and revise all equations in the article to ensure consistent formatting.
Authors answer 2.7
The format of all equations has been reviewed and corrected. The variables that merit it are in regular font.
Comment 2.8
Most of the references cited in this paper are old and fail to reflect the cutting-edge of the research. Please update and supplement the references.
Authors answer 2.8
To improve the manuscript, the next references were included.
Xipeng Pan, Lingqiao Li, Huihua Yang, Zhenbing Liu, Jinxin Yang, Lingling Zhao, Yongxian Fan, Accurate segmentation of nuclei in pathological images via sparse reconstruction and deep convolutional networks, Neurocomputing, Volume 229, 2017, Pages 88-99, ISSN 0925-2312, https://doi.org/10.1016/j.neucom.2016.08.103.
Zhang W, Sun X, Zhou L, Xie X, Zhao W, Liang Z, Zhuang P. Dual-branch collaborative learning network for crop disease identification. Front Plant Sci. 2023 Feb 10;14:1117478. doi: 10.3389/fpls.2023.1117478. PMID: 36844059; PMCID: PMC9950499
Xipeng Pan, Jijun Cheng, Feihu Hou, Rushi Lan, Cheng Lu, Lingqiao Li, Zhengyun Feng, Huadeng Wang, Changhong Liang, Zhenbing Liu, Xin Chen, Chu Han, Zaiyi Liu, SMILE: Cost-sensitive multi-task learning for nuclear segmentation and classification with imbalanced annotations, Medical Image Analysis, Volume 88, 2023, 102867, ISSN 1361-8415, https://doi.org/10.1016/j.media.2023.102867.
Zhang, Z. Li, H. -H. Sun, Q. Zhang, P. Zhuang and C. Li, "SSTNet: Spatial, Spectral, and Texture Aware Attention Network Using Hyperspectral Image for Corn Variety Identification," in IEEE Geoscience and Remote Sensing Letters, vol. 19, pp. 1-5, 2022, Art no. 5514205, doi: 10.1109/LGRS.2022.3225215.
Liu, X.; Song, L.; Liu, S.; Zhang, Y. A Review of Deep-Learning-Based Medical Image Segmentation Methods. Sustainability 2021, 13, 1224. https://doi.org/10.3390/su13031224
Sanchez-Patiño, A. Toriz-Vazquez, N. Hevia-Montiel and J. Perez-Gonzalez, "Convolutional Neural Networks for Chagas’ Parasite Detection in Histopathological Images," 2021 43rd Annual International Conference of the IEEE Engineering in Medicine & Biology Society (EMBC), Mexico, 2021, pp. 2732-2735, doi: 10.1109/EMBC46164.2021.9629563.
Putzu, L.; Fumera, G. An Empirical Evaluation of Nuclei Segmentation from H&E Images in a Real Application Scenario. Appl. Sci. 2020, 10, 7982. https://doi.org/10.3390/app10227982
O. Vuola, S. U. Akram and J. Kannala, "Mask-RCNN and U-Net Ensembled for Nuclei Segmentation," 2019 IEEE 16th International Symposium on Biomedical Imaging (ISBI 2019), Venice, Italy, 2019, pp. 208-212, doi: 10.1109/ISBI.2019.8759574.
Round 2
Reviewer 1 Report
The authors answered all my questions, and I'd like to thank them for that.
Author Response
We appreciate your comments, we have already addressed the suggested changes to improve the manuscript.
Reviewer 2 Report
The authors have well addressed all my concerns in the revision. The manuscript is acceptable for publication in its present form.
Author Response

(The authors gave the same response as above.)
